# Text classification to streamline online wildlife trade analyses

**Oliver C. Stringham** [1,2]*, **Stephanie Moncayo**[1], **Katherine G. W. Hill**[1], **Adam Toomes**[1], **Lewis Mitchell**[2], **Joshua V. Ross**[2], **Phillip Cassey**[1]

**1** Invasion Science & Wildlife Ecology Lab, University of Adelaide, Adelaide, SA, Australia, **2** School of Mathematical Sciences, University of Adelaide, Adelaide, SA, Australia

* oliverstringham@gmail.com

**Data Availability Statement:** Data and code for text classification are available from the figshare repository at https://doi.org/10.6084/m9.figshare.14032742 and from GitHub at https://github.com/ocstringham/text_classification_wildlife_trade/.

## Abstract

Automated monitoring of websites that trade wildlife is increasingly necessary to inform conservation and biosecurity efforts. However, e-commerce and wildlife trading websites can contain a vast number of advertisements, an unknown proportion of which may be irrelevant to researchers and practitioners. Given that many wildlife-trade advertisements have an unstructured text format, automated identification of relevant listings has not traditionally been possible, nor attempted. Other scientific disciplines have solved similar problems using machine learning and natural language processing models, such as text classifiers. Here, we test the ability of a suite of text classifiers to extract relevant advertisements from wildlife trade occurring on the Internet. We collected data from an Australian classifieds website where people can post advertisements of their pet birds (n = 16.5k advertisements). We found that text classifiers can predict, with a high degree of accuracy, which listings are relevant (ROC AUC ≥ 0.98, F1 score ≥ 0.77). Furthermore, in an attempt to answer the question 'how much data is required to have an adequately performing model?', we conducted a sensitivity analysis by simulating decreases in sample sizes to measure the subsequent change in model performance. From our sensitivity analysis, we found that text classifiers required a minimum sample size of 33% (c. 5.5k listings) to accurately identify relevant listings (for our dataset), providing a reference point for future applications of this sort. Our results suggest that text classification is a viable tool that can be applied to the online trade of wildlife to reduce time dedicated to data cleaning. However, the success of text classifiers will vary depending on the advertisements and websites, and will therefore be context dependent. Further work to integrate other machine learning tools, such as image classification, may provide better predictive abilities in the context of streamlining data processing for wildlife trade related online data.

## Introduction

The global wildlife trade is a major concern for biodiversity conservation and biosecurity enforcement [1]. Information on the composition and volume of species, and where they are traded, is highly valuable for informing conservation research and practice [2]. The Internet is

**Funding:** This research was funded by the Centre for Invasive Species Solutions (https://invasives.com.au/) Grant Number: PO1-I-002. The funders did not play any role in the study design, data collection and analysis, decision to publish, or preparation of the manuscript.

**Competing interests:** The authors have declared that no competing interests exist.

an emerging source of data on the wildlife trade [3,4]. Researchers, NGOs, and government agencies monitor websites that trade wildlife to quantify various aspects of the trade (e.g., [5]). Data gathered from the Internet are typically not immediately ready for analysis (i.e., they are 'messy') and must be cleaned or processed to identify the desired attributes for subsequent analysis [6]. This is especially true for classifieds, forums, and social media sites where human users type their advertisements into an open (or 'free form') text box. Consequently, relevant attributes cannot be extracted automatically (i.e., through web scraping or computer-based data manipulation) due to non-uniformity across users' advertisements (different species names, abbreviations, misspelling, etc.) [7]. Likewise, depending on the website, many online listings (i.e., posts) may contain items or taxa that are irrelevant for a given research context. For instance, in a pet reptile forum, one can find users trading tanks, food, or other accessories, which may not be relevant to researchers exploring the trade of live reptiles (e.g., [8]). The most common method to extract online wildlife trade data is to manually inspect each listing and record the desired attributes. Depending on how many listings are collected, the data cleaning process could represent an enormous amount of time and effort for researchers. Wildlife-related web data is notorious for its scale: for example, Xu et al. [9] tracked around 140k tweets from a two-week period relevant to ivory and pangolin trade.

Automated methods of data cleaning such as machine-learning techniques and Natural Language Processing (NLP) tools have potential to streamline the processing of wildlife trade data derived from the Internet [10]. A useful but unexplored application is to predict and extract relevant online listings based on their text, which could save time in manual data processing steps if many irrelevant listings exist in the dataset. In particular, text classification models relate the words associated with a particular label, such as 'relevant' or 'irrelevant', to predict the label of an unknown data point. A well-known application of text classifiers is filtering spam emails [11]. In this context, a text classification model uses a training dataset of labelled emails (spam or not spam) and trains a model to predict those labels based on their constituent words. The resulting model labels new incoming emails as spam or not. In the context of wildlife-trade data derived from the Internet, text classification models have the potential to identify relevant listings and remove irrelevant listings that do not sell wildlife (i.e., fish tanks, bird cages, food) by using the words in the listings. If shown to be effective, this could save researchers substantial time in the data cleaning process.

Here, we examine the efficacy of text classification models in predicting the relevance of wildlife trade advertisements on the Internet. By "relevance", we mean "pertaining to the wildlife trade"–filtering out advertisements for e.g., tanks, cages, food, and retaining only genuine advertisements for wildlife. Further, to assist future implementation of such models, we sought to identify how much data is needed for a text-classification model to perform adequately well. We collected advertisement listings from a popular Australian classifieds website where people trade their pet birds and accessories (e.g., bird cages or bird toys). Bird trade is largely unregulated in Australia (but see [12]) and is highly diverse with a large number of both native and alien species; with potential conservation and biosecurity consequences ([13]). We observed three major categories of advertisements that were irrelevant to our research objectives: (i) 'junk' listings (not trading birds); (ii) wanted advertisements (requesting a bird); and (iii) the sale of domestic poultry–e.g., gamebirds, waterfowl, and pigeons (non-target wildlife taxa). We manually labelled around 16.5k listings and tested the efficacy of three commonly used text classification models at determining which listings were relevant versus irrelevant. Next, we systematically removed records from our dataset and recorded the change in model performance. Our results imply that text classification can be an incredibly useful time-saver when cleaning data on the wildlife trade, which is structurally (textually) similar to the data we explore here.

## Materials and methods

### Data collection and data curation

We collected data from a popular Australian classifieds website daily over the course of five months (5 July 2019 to 5 December 2019). All information collected from the website was publicly available. We received ethics approval from The University of Adelaide Human Research Ethics Committee (approval number: H-2020-184) to collect this data and have anonymized the name of the website as good ethical practice [14]. On the website, people can post advertisements (i.e., listings) of items/animals they are trading. From each listing, we collected: (i) the title; (ii) text description; (iii) date; and, (iv) images (if provided). The title and text description fields are open text boxes where the user can type whatever they desire up to a character limit. We collected a total of 66,704 unique listings. Given the large number of unique listings collected, and the substantial resources required to manually clean the data, we labelled a random subset of around 25% of the listings (n = 16,509). This took approximately 103 hours to label (at an average rate of 161 listings per hour). Four different authors were labelers (SM, KH, AT, OS), and we did not overlap labelling, although this is preferred practice (e.g. see [15]).

For each listing, we manually labelled the taxa (e.g., species) being traded based on the title, the text, and the pictures provided in the listing. Some listings contained more than one species being traded. We identified the listing to the most specific taxonomic rank as possible (species or subspecies), but occasionally not enough information was provided and the listing was identified to genus, family, order, or class (i.e., bird). We resolved taxa names and obtained upper-level taxonomy using the Global Biodiversity Information Facility database [16]. For each listing, we recorded if the user was requesting a bird species (i.e., a wanted advertisement), except in the case of domestic poultry species (see below). We labelled listings not trading a live bird as 'junk' (i.e., bird accessory such as cage or bird food).

### Preparing text for text classification models

We considered all text written by the user (title and text description) for our analyses. To prepare or 'clean' the text for the NLP text classification models, we followed standard NLP text cleaning procedures [17] and removed special characters (emojis, dollar signs, numbers, etc.), removed all punctuation, converted text to lowercase, and removed all numbers. Next, we removed all stop words found in the following lexicons: *SMART*, *snowball*, and *onix*. We did not remove the stop words: "want", "wants", "wanting", or "wanted", so we could distinguish wanted advertisements. We stemmed each word using the Snowball stemmer. For the text classifier models, we tokenized the text to be unigrams (i.e., one word) and did not consider further n-grams. We encoded the unigrams as count vectors and used these vectors as features for the text classification models. Text cleaning was performed in the statistical software R (version 3.6.3) [18] using the following packages: stringr (version 1.4.0) [19], dplyr (version 1.0.5) [20], tidytext (version 0.2.3) [17], and corpus (version 0.10.1) [21].

To test the classification of irrelevant listings (see '*Text classification models*' below), we applied three separate labels for each listing. The first label was for 'junk' listings, where a live bird was not being traded (e.g., bird cage). The second label was for 'wanted' listings where a user was requesting a bird species and not selling one. The final label was for taxa that we considered non-target for our purposes (i.e., farm, poultry, or domesticated species). We called this label 'domestic poultry' and applied it to listings that were selling birds in the taxonomic orders of Anseriformes (waterfowl) and Galliformes (gamebirds) or trading domestic pigeons (*Columba livia domestica*). For text classification models, we removed listings categorized as

more than one label (i.e., 'domestic poultry' and 'wanted'). Further, for the 'wanted' label, we removed listings if eggs were being advertised, as we did not simultaneously record if egg advertisements were also labeled as 'wanted'. This resulted in a sample size (number of listings used for text classification models) of 16,475 for 'domestic poultry', 16,446 for 'junk', and 13,751 for 'wanted'.

## Text classification models

To classify irrelevant listings, we used three common supervised text classifiers: Logistic Regression, Multinomial Naive Bayes, and Random Forest. At a basic level, each classifier considers each word (i.e., gram) and their frequency as a covariate (i.e., 'feature') [22]. However, each classifier varies in the algorithm used to classify observed listings as relevant or not [22]. For each classifier, the order of the words in the listing was unaccounted, thus earning the name 'bag of words' classifier. We ran each model for each of the three labels mentioned above. We used 10-fold cross validation to train the model and evaluate predictions. We used the cross-validated macro-average of the following metrics to evaluate the performance of each model: receiver operating characteristic (ROC) curve and its area under the curve (ROC AUC), precision-recall curve and its area under the curve (PR ROC), precision, recall, negative predictive value (NPV), specificity, and F1 score (see S1 Appendix for more information evaluation metrics). We extracted the top features (e.g., covariates) for each model. Text classification models were performed in Python using the sci-kit learn library (version 0.23.2) [23], while plotting was conducted in R using ggplot2 (version 3.3.3) [24].

## Sensitivity analysis: Degradation of model performance with diminishing sample size

To test the sensitivity of model performance to changes in sample size, we implemented the text classification model with iteratively smaller sample sizes. We systematically decreased the sample size of the training set by 500 records at a time, removing at most 15k records (c. 91% of entire dataset). We repeated this for each label and used 10-fold cross validation. To account for the variability in model performance due to cross-validation, we repeated the text classification model for 100 iterations, for each sample size explored. We recorded 10-fold cross validation statistics across each fold and model iteration (1,000 values in total for each sample size). For this sensitivity analysis, we only considered the logistic regression classifier and used the F1 value to evaluate model performance. We recorded the maximum training set sample size at which the F1 score was 99% of its maximum value (i.e., the F1 score without reducing sample size).

## Results

We manually categorized 16,509 listings, of which 15.0% (n = 2,473) were labeled as 'junk', 21.9% (n = 3,615) were labeled as 'domestic poultry', 4.8% (n = 787) were labelled as 'wanted' advertisements, and the remaining (c. 58%) were 'for sale' advertisements of relevant bird taxa.

The text classifiers performed extremely well for the 'domestic poultry' label (Fig 1 and S2 Appendix), with a cross-validated average ROC AUC of >0.99, Precision-Recall AUC of ≥0.97, and F1 score of >0.95 for all text classifiers (Figs 1–3). The text classifiers for the 'junk' label also performed very well, with marginally lower metric values compared to 'domestic poultry' (Fig 1). Further, all other metrics evaluated suggested that the text classification models performed very well for these two labels (Figs 1–3 and S2 Appendix; see S3 Appendix for confusion matrices). The text classification models for the 'wanted' advertisement label

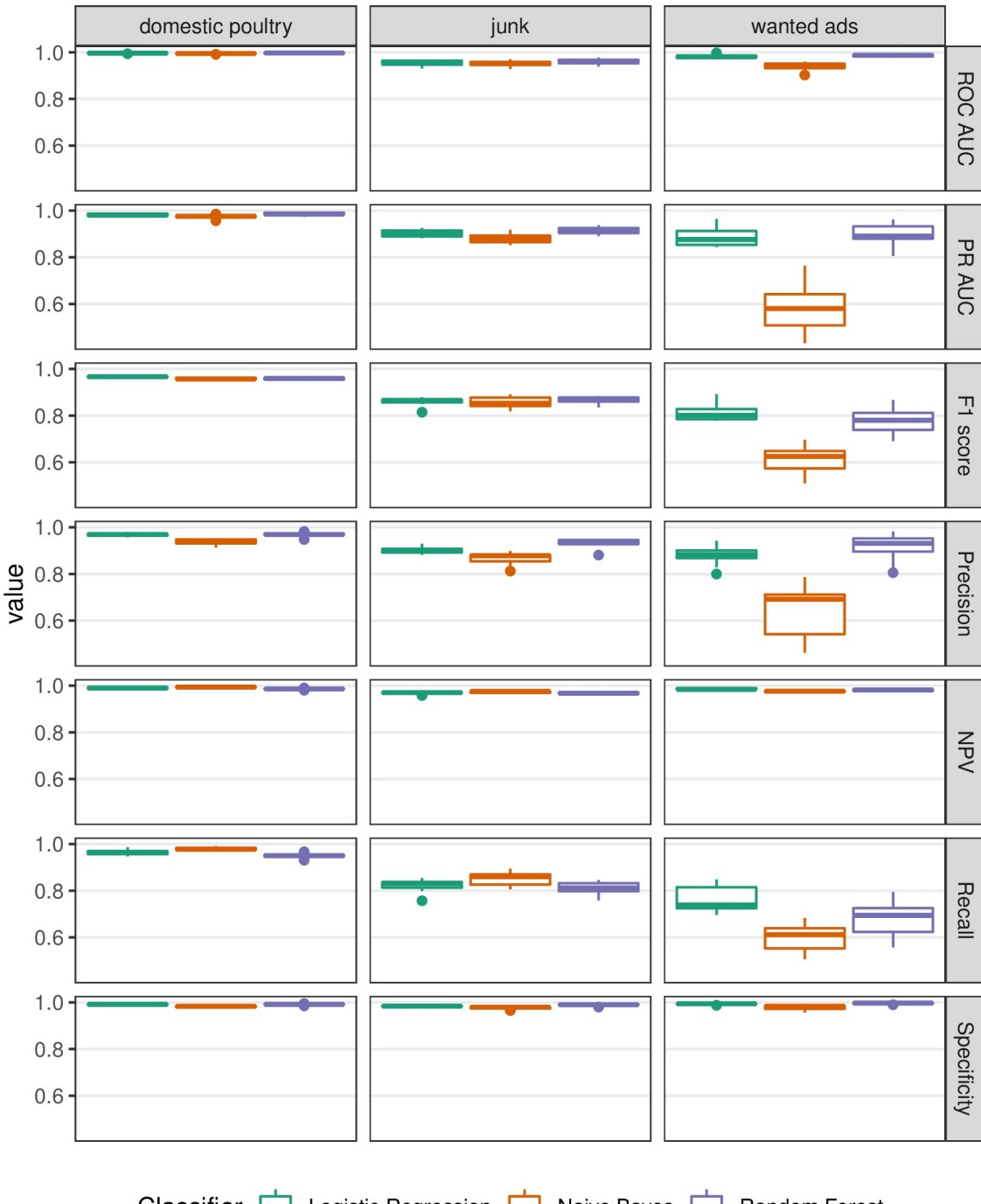

**Fig 1. Model evaluation metrics for text classifiers.** Evaluation metrics (rows) are derived from 10 cross-validation folds using different text classifiers evaluated for three different labels (columns). See S1 Appendix for more information and calculation of the evaluation metrics and S2 Appendix for exact metric values.

performed less well, however, the Logistic Regression and Random Forest classifiers for this label performed moderately well and each was much better than chance with a ROC AUC > 0.98, Precision-Recall AUC > 0.88, and F1 score > 0.77. Overall, the 'wanted'

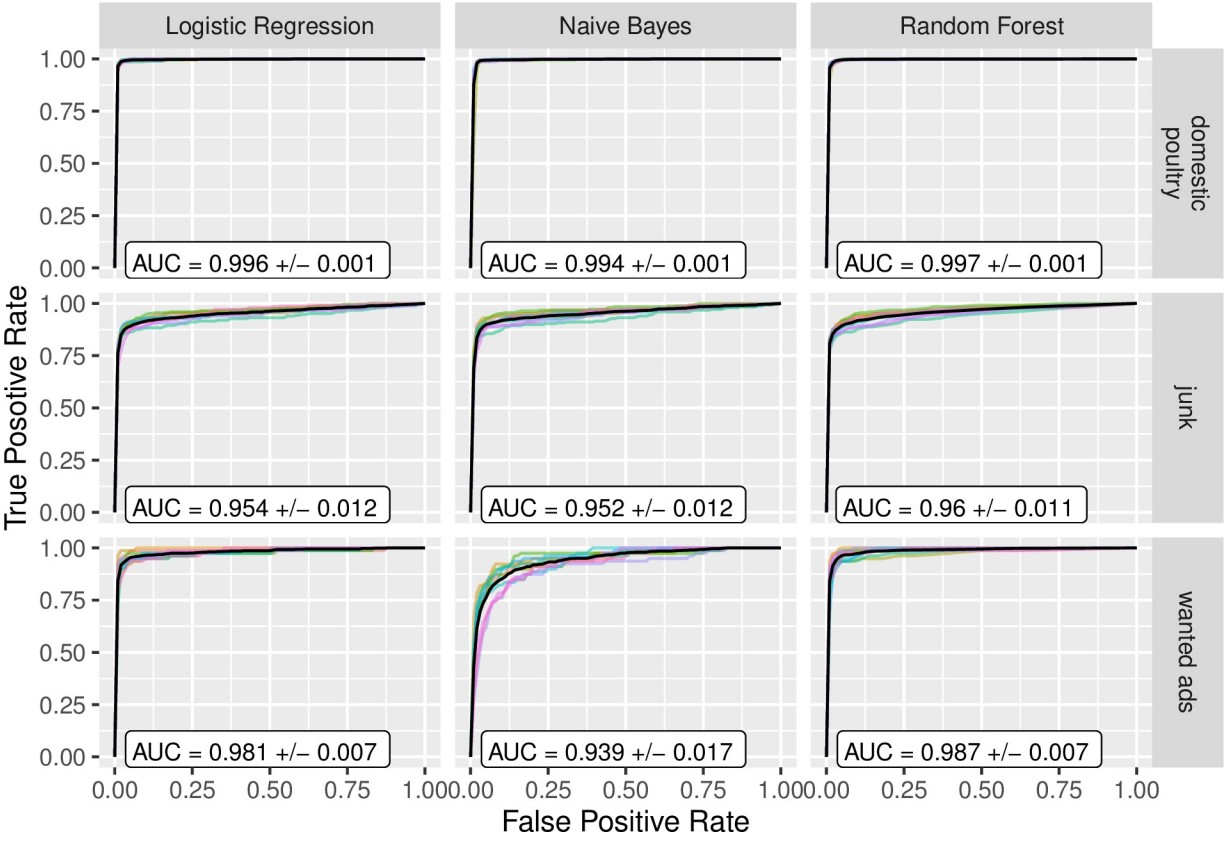

**Fig 2. Receiver operating characteristic curves and the area under the curve (ROC AUC).** Three different text classifiers (columns) were tested across three different labels (rows). For each panel, each line represents one cross-validation fold and the solid black line represents the average across all cross-validation folds. Average AUC (area under curve) values are reported with standard deviation.

classifiers were not as good at predicting positive outcomes (e.g., if a listing is 'wanted'), yet did not struggle with predicting negative outcomes (Specificity = 0.99, and Negative Predictive Value = 0.99 for Logistic Regression classifier). In terms of relative performance between the classifiers, the Logistic Regression and Random Forest classifiers slightly outperformed the Naive Bayes Classifier; however, overall, their performances were comparable (Figs 1–3).

The top features for each label aligned with what should be expected and were similar across all text classifiers (Fig 4). For the 'junk' label, grams such as "condit" (i.e., condition), "cage", "birdcag" (i.e., birdcage) were the top features. For the 'domestic poultry' label, grams such as "pigeon", "rooster", and "chicken" were the top features. Finally, for the 'wanted' label, grams such as "want", "buy", "wtb" (an acronym for 'want to buy'), and "unwant" (i.e. "unwanted") were the top features.

As we reduced the sample size of the training set, we observed a non-linear decrease in model performance, where the F1 score initially declined gradually and then at an increasing rate at lower sample sizes (Fig 5). There were differences in this decline in performance among labels. The classifier for the 'domestic poultry' label realized 99% of the full model F1 score at c. 4.8k records (29% of dataset). For the 'junk' label, this was c. 9.3k records and c. 6.3k records for the 'wanted' label (57% and 45%, respectively). Stated another way, for the 'domestic poultry' label, the addition of c. 11k labelled records from our manual data labelling only increased the model F1 score by 0.01. For 'junk' and 'wanted', this value was c. 7.1k listings and c. 7.5k listings, respectively.

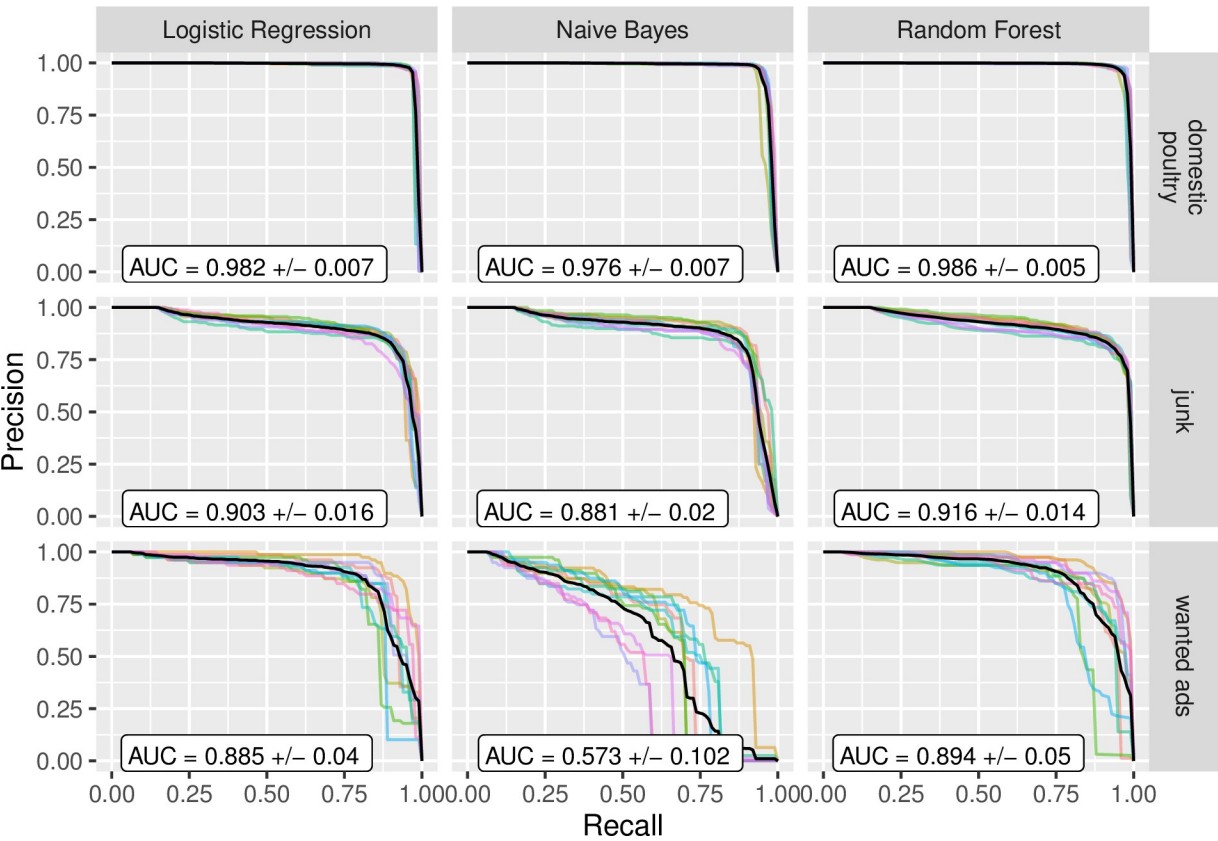

**Fig 3. Precision recall curves and the area under the curve (PR AUC).** Three different text classifiers (columns) were tested across three different labels (rows). For each panel, each line represents one cross-validation fold and the solid black line represents the average across all cross-validation folds. Average AUC (area under curve) values are reported with standard deviation.

## Discussion

Text classification can be a highly accurate method to extract relevant listings of wildlife found on the Internet. In particular, for listings trading non-target taxa and listings trading bird accessories (e.g., bird cages), text classification models were able to classify these listings with a very high degree of accuracy. Although the performance of the model varied between labels, our results suggest that this technique can be used to substantially lower the number of wildlife listings needed to be manually inspected, thus saving considerable time and resources. Further, we provide clarity around the question of 'how much data is needed to guarantee an adequately performing model?'. Of the more than 16k listings we manually labelled, our results suggest that, at most, only 9k listings were needed, although this number varied by label.

Text classification models are commonplace in other disciplines and industries, which work heavily with text data (e.g., [11]), yet have not been applied to data collected on the wildlife trade occurring on the Internet. Importantly, from our dataset, around 60% of the listings were relevant (for our purposes), representing a substantial amount of time and effort that would otherwise be spent on manually removing irrelevant data. For the website we explored, we showed that text classifiers predicted with great accuracy the advertisements that were not selling wildlife or selling non-target wildlife. In particular, text classification models performed the best for identifying listings trading non-target taxa (e.g., farm and domestic bird species). This is promising as a time saving tool because sometimes the most commonly traded taxa are

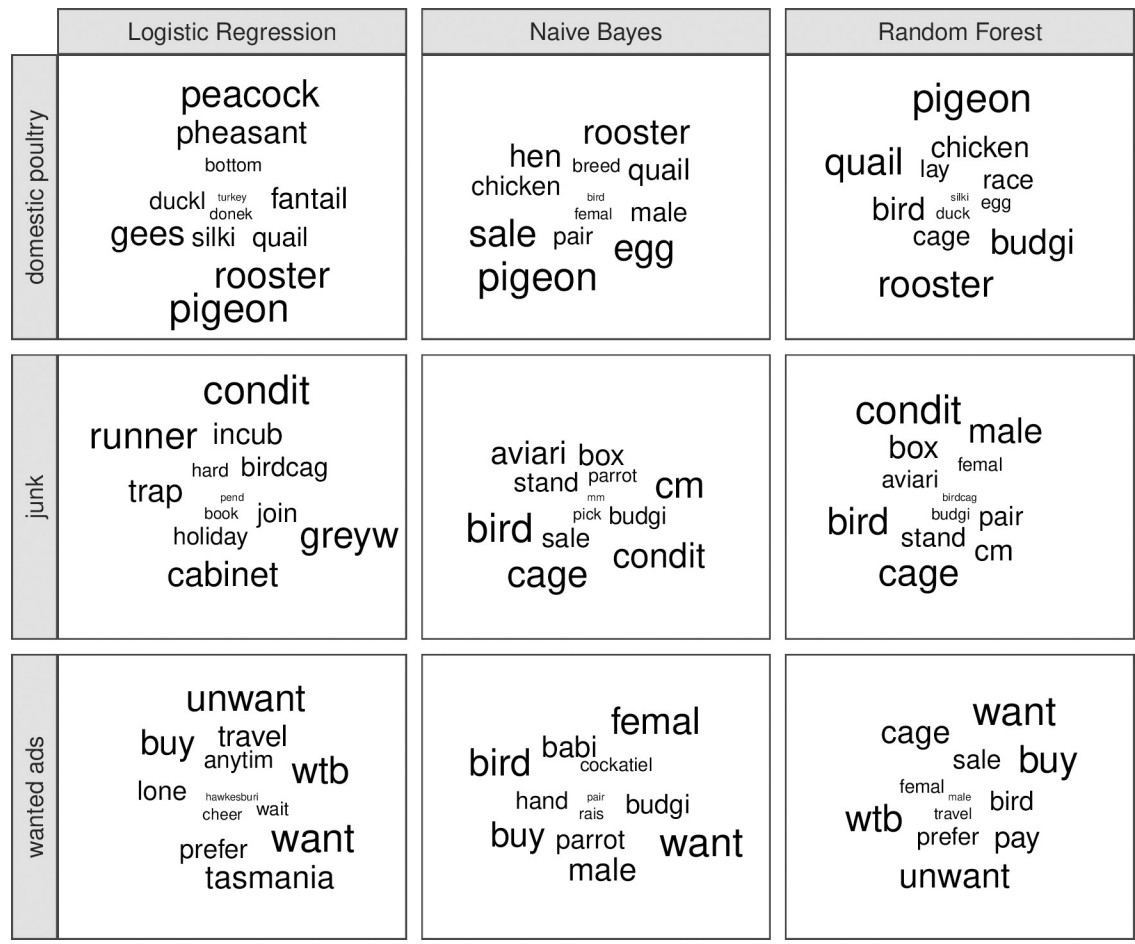

**Fig 4. Word clouds of top features of text classifiers.** Top words (i.e., features or grams) shown for each label (rows) and classifier (columns). The size of the word corresponds to importance, where larger words indicate higher importance. Note that words are stemmed (e.g., condition is stemmed to condit).

the ones of least interest to researchers (i.e., pigeons and chickens; in our example). In contrast, the text classifiers had more difficulty distinguishing 'wanted' advertisements (where a user was requesting a bird) yet was still much better than chance. This suggests that the words people used in wanted advertisements have some overlap with those words used in non-wanted advertisements (e.g., names of species), and thus yields lower predictive abilities. Importantly, we demonstrate that the model performance will likely not improve with more data because we observed a plateau of model performance after around 6k listings for wanted advertisements (45% of sample size). Therefore, even if we manually labelled many more listings, the model performance is unlikely to increase. This highlights an important point that model performance is a function of the underlying data itself (i.e., text) and not of the lack of data (once an adequate sample size is achieved).

How much data is required for an adequately performing text classification model? Our results show that this number will vary by what is being classified. For this study, we cleaned a substantial number of listings (c. 16k) yet found that model performance marginally increased after 5k to 9k records (31% to 56% of total effort). Thus, for other researchers who may not have the resources to invest this much effort, or are looking for a more efficient way to curate messy online data, our results provide guidance on how much data is needed before text

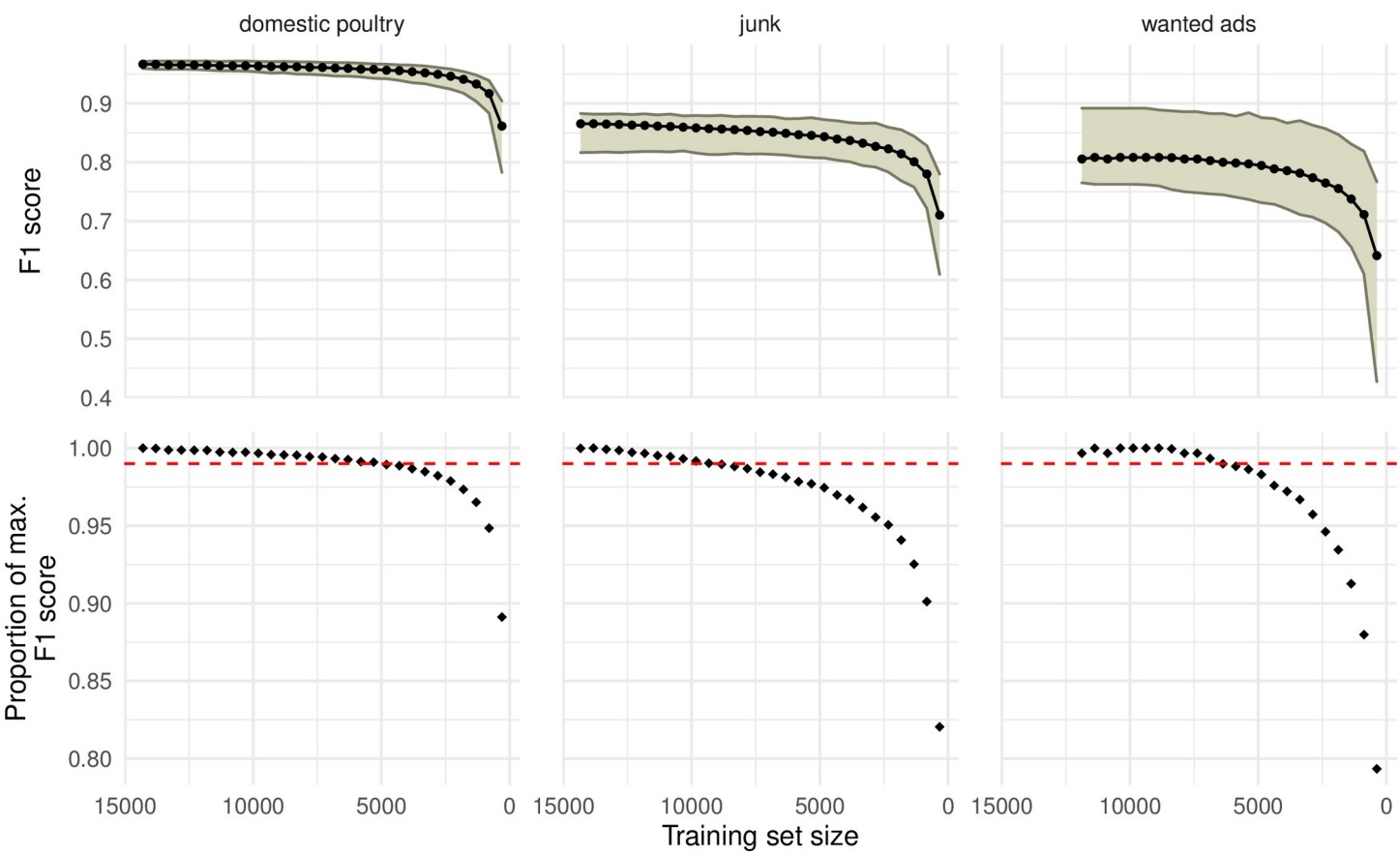

**Fig 5. The effects of reducing sample size on text-classifier model performance.** Top row: The F1 score evaluated at decreasing sample size (training set) values. Ribbons represent the 95% quantile range from 100 iterations of 10-fold cross validation logistic regression text classification, repeated for each specified label ('domestic poultry', 'junk', and 'wanted'). Bottom row: The proportion of the maximum F1 score, evaluated at each sample size, for each label. Only the median value was considered. The red horizontal line represents 0.99 of the maximum F1 score.

classification can be used. We recommend establishing computer code to test the model performance and then repeatedly check the model performance at regular intervals (e.g., every 1k records cleaned). Ultimately, the labelled dataset will need to encapsulate the variation of words (i.e., vocabulary) used for a particular label for the text classifier to perform well [22]. For instance, for the 'junk' label, the model performance plateaued at around 5k more records than it did for other labels. We hypothesize the words that Internet users write for the listings that fall under the 'junk' label has more variation (i.e., more words) and thus, we needed a larger sample size of labelled listings to account for that variation.

An important limitation of text classification (and other machine learning tools) is that they are highly context dependent [25]. Our specific classifiers were developed based on the text of birds being traded online in Australia and will likely be less useful for birds being traded in other countries and almost entirely useless if looking at other taxa (e.g., fish or plants) or in another language. The reason for this lack of generalization is because words used, and their frequency, will vary under different contexts. For instance, when looking at the trade of aquarium fish, a common irrelevant advertisement may be the sale of a fish tank, something that is not found when trading birds. We recommend that researchers consider each context separately when using these tools. Since manual data processing is likely always required to analyze the data, these tools can be tested throughout the cleaning stage to see if applicable.

Besides extracting relevant advertisements, text classifiers have the potential to identify the species being traded in online advertisements. Our results suggest that this will be possible for commonly traded taxa, with large amounts of data. For instance, in our study, advertisements for a group of species (waterfowl, gamebirds, and pigeons) comprised around 3.6k listings (22% of dataset) and were highly distinguishable using the text classifiers. The same kinds of models can be used to identify individual species of interest; however, text classifiers (like all machine learning techniques) require a large volume of data to perform well [10]. In many cases, individual species of interest may not have enough advertisements to build adequate text classifiers. Thus, alternative methods such as matching species names (scientific, common, or trade names) to the text of advertisements using a fuzzy string-matching model (e.g., Levenshtein distance) may yield better results. In fact, if consistent patterns are used by users (e.g., the same species name is used by many users), string matching may yield just as good or better results than text classifiers. While our study relied exclusively on the text of the advertisement, there are other attributes of an Internet listing that can be considered for automated cleaning. For instance, a related study used metadata attributes of online listings (e.g., the number of views and the price) to classify illegal sales of elephant ivory [26]. In cases with no or limited text provided (e.g., only a photo is posted), machine learning techniques such as image classification could assist in the classification of species or the product traded [27]. Integrating text classification with the aforementioned models may improve predictive ability, and we recommend this as a future area of research and development for the wildlife trade related online data.

Given that a substantial proportion of online listings may not be relevant to wildlife trade research (e.g., 40% irrelevant for our dataset), text classification methods can substantially decrease the amount of time spent processing raw data. Here, we demonstrate that text classification can be viable tool to identify irrelevant listings. When considering data on the scale of 'big data' of tens to hundreds of thousands of online advertisements (e.g., [28]), text classifiers have the potential to save tens to hundreds of hours of curation effort. We recommend future application of text classifiers and testing other machine learning and natural language processing tools when cleaning messy data collected from the Internet on wildlife trade.

## Supporting information

**S1 Appendix. Definitions of metrics used.**
(DOCX)

**S2 Appendix. Table of model metrics.**
(DOCX)

**S3 Appendix. Confusion "matrices".**
(DOCX)

## Acknowledgments

The authors acknowledge the Indigenous Traditional Owners of the land on which the University of Adelaide is built -the Kaurna people of the Adelaide Plains.

## Author Contributions

**Conceptualization:** Oliver C. Stringham, Lewis Mitchell, Joshua V. Ross, Phillip Cassey.

**Data curation:** Oliver C. Stringham, Stephanie Moncayo, Katherine G. W. Hill, Adam Toomes.

**Formal analysis:** Oliver C. Stringham.

**Funding acquisition:** Lewis Mitchell, Joshua V. Ross, Phillip Cassey.

**Investigation:** Oliver C. Stringham.

**Methodology:** Oliver C. Stringham.

**Supervision:** Oliver C. Stringham, Lewis Mitchell, Joshua V. Ross, Phillip Cassey.

**Visualization:** Oliver C. Stringham.

**Writing – original draft:** Oliver C. Stringham.

**Writing – review & editing:** Oliver C. Stringham, Stephanie Moncayo, Katherine G. W. Hill, Adam Toomes, Lewis Mitchell, Joshua V. Ross, Phillip Cassey.

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
