## [Decision Letter · Decision Letter 0]

7 Apr 2021

PONE-D-21-05869

Text classification to streamline online wildlife trade analyses

PLOS ONE

Dear Dr. Stringham,

Thank you for submitting your manuscript to PLOS ONE. After careful consideration, we feel that it has merit but does not fully meet PLOS ONE’s publication criteria as it currently stands. Therefore, we invite you to submit a revised version of the manuscript that addresses the points raised during the review process.

ACADEMIC EDITOR:

The reviewers have asked for revisions. Some of the reviewers have questioned the novelty of the paper. There is also concerns about the discussion and the comparisons, that authors need to address. Based on all this, I am recommending major revisions. 

Furthermore when submitting the revised paper, please also consider the following points:

1. English language needs proofreading.

2. References should be in proper format.

3. All acronyms must first be defined.

We look forward to receiving your revised manuscript.

Kind regards,

Usman Qamar

Academic Editor

PLOS ONE

Journal Requirements:

Additional Editor Comments (if provided):

Reviewers' comments:

Reviewer's Responses to Questions

**Comments to the Author**

1. Is the manuscript technically sound, and do the data support the conclusions?

Reviewer #1: Partly

2. Has the statistical analysis been performed appropriately and rigorously? 

Reviewer #1: Yes

3. Have the authors made all data underlying the findings in their manuscript fully available?

Reviewer #1: Yes

4. Is the manuscript presented in an intelligible fashion and written in standard English?

Reviewer #1: Yes

5. Review Comments to the Author

Reviewer #1: PLOS ONE

Manuscript Number: PONE-D-21-05869

Text classification to streamline online wildlife trade analyses

** The text is not really specific in explaining what is the problem that you are trying to address:

“Here, we examine if text classification models can predict which Internet listings are relevant to wildlife 72 trade research (for our own specific research purposes; e.g., [12]).”

I think you should be more specific.

** I think the domain (application) is very narrow.

** The (87 Data collection and data curation) section starting line 87 lacks clarity. You manually labeled the advertisement, what are the labels? Did you label them as ‘wanted’, ‘for sale’, ….etc. or as type/species ….etc. what are the species. So many unknowns.

**how the word features were encoded? Did you use tfidf? Or what?

** in the table (Appendix S3: Confusion “matrices”) the numbers do not match (total 361 vs 362)? Why?

Logistic Regression 348 13

Naive Bayes 354 8

Random Forest 344 18

** Also for label junk, the numbers are wrong 246 vs 247? Did you review the paper before submitting?

** Overall, the paper is written fairly well (however, more language revision can improve it even more). Several points are not clear in the paper (some are mentioned above in this review). The main problem (specific problem) is not clearly stated in this paper. From my reading of this paper I can see that this work is a simple and straightforward text classification task with three already well established text classifier (learners) and with the standard text preprocessing steps (stemming, stop-word removal,…etc.); and with standard bag-of-word (feature encoding is not explained in the paper though). With all these, what is the main contribution of this work? I can see that one of the main contributions of this work is the type of problem (wildlife trading, maybe I am note sure.). I can see that there is nothing new (nothing unique) about using ML or text classification in this paper.

6. PLOS authors have the option to publish the peer review history of their article (what does this mean?). If published, this will include your full peer review and any attached files.

Reviewer #1: No

---

## [Author Response · Author response to Decision Letter 0]

25 May 2021

Please refer to the ‘Response to Reviewers’ document for detailed response to each peer-reviewed comment.

---

## [Decision Letter · Decision Letter 1]

4 Jun 2021

PONE-D-21-05869R1

Text classification to streamline online wildlife trade analyses

PLOS ONE

Dear Dr. Stringham,

Thank you for submitting your manuscript to PLOS ONE. After careful consideration, we feel that it has merit but does not fully meet PLOS ONE’s publication criteria as it currently stands. Therefore, we invite you to submit a revised version of the manuscript that addresses the points raised during the review process.

We look forward to receiving your revised manuscript.

Kind regards,

Usman Qamar

Academic Editor

PLOS ONE

Journal Requirements:

Reviewers' comments:

Reviewer's Responses to Questions

**Comments to the Author**

1. If the authors have adequately addressed your comments raised in a previous round of review and you feel that this manuscript is now acceptable for publication, you may indicate that here to bypass the “Comments to the Author” section, enter your conflict of interest statement in the “Confidential to Editor” section, and submit your "Accept" recommendation.

Reviewer #1: All comments have been addressed

2. Is the manuscript technically sound, and do the data support the conclusions?

Reviewer #1: Partly

3. Has the statistical analysis been performed appropriately and rigorously? 

Reviewer #1: I Don't Know

4. Have the authors made all data underlying the findings in their manuscript fully available?

Reviewer #1: Yes

5. Is the manuscript presented in an intelligible fashion and written in standard English?

Reviewer #1: Yes

6. Review Comments to the Author

Reviewer #1: PLOS ONE Manuscript Number: PONE-D-21-05869R1

Text classification to streamline online wildlife trade analyses

I believe the authors have addressed some of the review comments but not all.

In my previous (first) review I mentioned:

“The main problem (specific) is not clearly stated in this paper. From my reading of this paper I can see that this work is a simple and straightforward text classification task with three already well established text classifier (learners) and with the standard text preprocessing steps (stemming, stop-word removal,…etc.); and with standard bag-of-word (feature encoding is not explained in the paper though). With all these, what is the main contribution of this work? I can see that one of the main contributions of this work is the type of problem (wildlife trading). I can see that there is nothing new (nothing unique) about using ML or text classification in this paper.”

I think this point is still not clear to me. In other words: What is the gain of this research in the wild-life trade application? You said “to streamline online wildlife…” what do you mean by ‘to streamline”?

The revisions done on this paper/version are very small (very limited): only a few lines around line 72 and only few lines around line 120; that is it!

7. PLOS authors have the option to publish the peer review history of their article (what does this mean?). If published, this will include your full peer review and any attached files.

Reviewer #1: No

---

## [Author Response · Author response to Decision Letter 1]

10 Jun 2021

Please see Cover Letter and Response to Reviewer documents for our responses to specific reviewer and editor comments.

---

## [Editor Report · Decision Letter 2]

18 Jun 2021

Text classification to streamline online wildlife trade analyses

PONE-D-21-05869R2

Dear Dr. Stringham,

We’re pleased to inform you that your manuscript has been judged scientifically suitable for publication and will be formally accepted for publication once it meets all outstanding technical requirements.

Kind regards,

Usman Qamar

Academic Editor

PLOS ONE
---

## [Editor Report · Acceptance letter]

23 Jun 2021

PONE-D-21-05869R2 

Text classification to streamline online wildlife trade analyses 

Dear Dr. Stringham:

I'm pleased to inform you that your manuscript has been deemed suitable for publication in PLOS ONE. Congratulations! Your manuscript is now with our production department. 

Kind regards, 

on behalf of

Dr. Usman Qamar 

Academic Editor

PLOS ONE